# High frequency head impact exposure changes hippocampal sharp-wave ripple architecture

Daniel P. Chapman[ID]1,2☯, Margaret S. Sten1,2☯, Stefano Vicini2,3, Mark P. Burns[ID]1,2*

1 Department of Neuroscience, Georgetown University Medical Center, Washington, District of Columbia, United States of America, 2 Interdisciplinary Program in Neuroscience, Georgetown University Medical Center, Washington, District of Columbia, United States of America, 3 Department of Pharmacology and Physiology, Georgetown University Medical Center, Washington, District of Columbia, United States of America

☯ These authors contributed equally to this work.
* mpb37@georgetown.edu

## Abstract

Repeated head impact in sports leads to chronic cognitive and neurobehavior deficits even in the absence of brain pathology. High-frequency head impact (HFHI) in mice causes a chronic change to the synaptic properties in core hippocampal circuits, and causes no cell death, no axonal damage, no tau or amyloid accumulation, and no inflammation, yet results in impaired cognitive function. It is unknown how HFHI affects intrinsic plasticity events and if neural biomarkers of HFHI can be detected. Sharp-wave ripples (SWR) are hippocampal population events consisting of a sharp wave (1–30 Hz) and associated ripple oscillation (120–220 Hz). SWR are strongly associated with memory and are an established biomarker for cognition and memory. To characterize the effect of HFHI on SWR, we prepared acute slices 24 hours after HFHI and used field recordings to characterize hippocampal SWR. Physiological SWR were present in both sham and HFHI mice, and their architecture was grossly intact. We did not detect pathological ripples. Quantification of SWR features showed a decreased amplitude and power of SWR in HFHI brains compared to sham. Further analysis of the ripple oscillations found decreased number of ripple cycles within each event, and reduced ripple power in HFHI brains. These data show that HFHI alters hippocampal SWR architecture, reducing metrics including SWR amplitude and power, which may play a role in impaired cognition in this mouse model.

## Introduction

Traumatic brain injury (TBI) is the most common neurological disorder in the world [1–3]. Repeated exposure to mild TBI during an athletic career has been linked to memory impairments and increased risk of neurodegenerative diseases such as chronic traumatic encephalopathy (CTE) [4–6]. The observable pathology character-istics of CTE of axonal injury, inflammation, and tau accumulation does not directly

**Data availability statement:** Data used in this study are available through the Open Data Commons for Traumatic Brain Injury (odc-tbi.org; RRID:SCR_021736), Chapman et al., (2025), https://doi.org/10.34945/F5QG7K.

**Funding:** Research Reported in this publication was supported by the National Institute of Neurological Disorders and Stroke of the National Institutes of Health under award numbers R01NS121316, R01NS107370 (MPB) & F30NS122281 (DPC). The research was also supported by the CTE Research Fund at Georgetown University (MPB). The content is solely the responsibility of the authors, and does not necessarily represent the official views of the National Institutes of Health or donors.

**Competing interests:** The authors have declared that no competing interests exist.

correlate to symptoms of repeat head impacts especially in the early stages of disease [7]. More recent data shows that even subclinical head impacts not captured by the current classification of a single incidence of TBI, like heading a soccer ball and sustained low-amplitude head movements faced by professional sled athletes, can cause chronic cognitive symptoms [8–10]. The mechanisms by which these sustained sub-concussive, pathology-negative head impacts lead to memory dysfunction is still largely unknown. Finding biomarkers that indicate abnormal brain function after sub-concussive head impacts is an important step in identifying and treating such injury.

Preclinical researchers have developed several animal models of repeat mild TBI with learning/memory deficits occurring in the absence of pathology [11–14]. We have characterized a model of high-frequency head impacts (HFHI) in which mice receive 30 head impacts over the course of one week, intended to emulate the number of head impacts sustained in a week of practice and play by a collegiate football athlete [15,16]. These mice display widespread deficits in anterograde and retrograde learning and memory which is associated with decreased long-term potentiation (LTP) in the CA1 region, decreased AMPA/NMDA ratio in CA1 neurons, decreased ensemble coordination in CA1 neurons, and synaptic deficits in the memory engram of the dentate gyrus (DG) all in the absence of neuron cell death, pathology, and inflammation [16–18]. In addition, the firing properties of CA1 neurons are transiently disturbed following HFHI however, these changes are short-lived and do not correlate with cognitive disruptions seen at chronic timepoints [16,19]. The development of HFHI-induced cognitive deficits can be blocked with pretreatment of the NMDA receptor antagonist memantine, and retrograde memory loss can be recovered through optogenetic stimulation of the DG memory engram, further supporting a synaptic mechanism, rather than pathological or inflammatory mechanisms, of cognitive deficit in this mouse model [16,18]. In sum, we have previously explored stimulated plasticity extensively in the HFHI model, but it is still unknown how synaptic changes affect intrinsic plasticity and whether established biomarkers of cognitive function remain intact in HFHI mice.

Hippocampal sharp wave ripples (SWR) are robust population events that occur during slow-wave sleep and consummatory behaviors and have been linked to memory consolidation and planning [20,21]. SWR are spontaneous events lasting between 30–150 ms and consist of a sharp wave, evaluated at lowpass frequencies (1–30 Hz), and co-occurring ripple events (120–220 Hz) [20]. SWR are disrupted in other known causes of memory dysfunction such as Alzheimer's disease [22] and stimulating, disrupting, or prolonging SWR has shown causal evidence for their involvement in memory [23–25]. Ripples are fast oscillations 120–220 Hz that occur in the CA1 region. They may occur with or without coinciding sharp wave events and it is thought that the depolarization that occurs during sharp waves opens a window for ripple events to occur [20,26,27]. In addition to their potential to be targeted through intervention [23], SWR also serve as a cognitive biomarker for learning and memory. Thus, we sought to characterize the effect of HFHI on hippocampal SWR in acute brain slices.

## Methods

### Animals and HFHI model

All procedures were performed in accordance with protocols approved by the Georgetown Animal Care and Use Committee (2016−1245). The closed head impact procedures were performed on male C57BL/6J mice (*Mus* musculus), as previously described [16,19]. Two to three-month-old mice were anesthetized using 3% isoflurane in 1.5 L/min oxygen for 2 minutes. Mice were then given an additional minute of isoflurane in the injury device with their unrestrained head resting on a gel pad. A teflon tip with a 10 mm diameter was positioned directly over the midline dorsal surface of the head with the front of the impact tip positioned immediately caudal to the eye socket and equidistance from the mouse ears. Pneumatically controlled impacts were delivered at an impact speed of 2.35 m/s, dwell time of 32 ms, and an impact depth of 7.5 mm. In total, 30 hits were given over 6 days with 5 hits given per day over a 10 second interval. Sham animals received identical handling and anesthesia protocols, but no head impacts.

### Slice preparation

Acute transverse hippocampal slices were prepared from experimental animals 24 hours after the last impact (described above). Brains were sliced and incubated in NMDG and HEPES-buffered artificial cerebrospinal fluid (aCSF), as previously described [16,18]. On the day of the experiments, one sham and one HFHI mouse were anesthetized in open isoflurane. Transcardial perfusion was performed using 4º C NMDG solution (92 mM NMDG, 2.5 mM KCl, 1.25 mM NaH2PO4•2H2O, 30 mM NaHCO3, 20 mM HEPES, 25 mM glucose, 10 mM sucrose, 5mM ascorbic acid, 2 mM thiourea, 3 mM sodium pyruvate, 5 mM N-acetyl-L-cysteine, 10 mM MgSO4•7H2O, 0.5 mM CaCl2•2H2O, pH~7.4, osmolarity~300–310 mOsm). Brain dissection was performed after perfusion and excised brains were sliced in 4º C NMDG solution. 500 μm thick transverse slices were prepared using a Vibratome Series 3000. Following slicing, hemisected brain slices were immediately placed in 32º C NMDG solution for 25 min. The gradient of sodium chloride concentration was increased up to 90 mM with sodium spike-ins every 5 minutes before slices were then transferred to an incubation chamber containing room temperature carboxygenated low-glucose HEPES solution (92 mM NaCl, 2.5 mM KCl, 1.25 mM NaH2PO4•2H2O, 30 mM NaHCO3, 20 mM HEPES, 2.5 mM glucose, 5 mM ascorbic acid, 2 mM thiourea, 3 mM sodium pyruvate, 5 mM N-acetyl-L-cysteine, 2 mM MgSO4•7H2O, 2 mM CaCl2•2H2O, pH~7.4, osmolarity~280–290 mOsm) and were allowed to recover until the recording.

### Electrophysiology

Following recovery, slices prepared as above were transferred to a Siskiyou PC-H perfusion chamber, anchored to the bottom of the recording chamber and submerged in carboxygenated aCSF (124 mM NaCl, 3.5 mM KCl, 1.2 mM NaH2PO4•2H2O, 26 mM NaHCO3, 10 mM glucose, 1 mM MgCl2•6H2O, 2 mM CaCl2•2H2O, pH~7.4, osmolarity~300–310 mOsm). The aCSF was heated to 30º C and perfused through the recording chamber at 5 mL/min. Recordings were performed with a Multiclamp 700B amplifier (Molecular Devices, San Jose, CA), digitized to 20 kHz, and band-pass filtered between 1 Hz - 1kHz with a computer running Clampex 11 and DigiData 1440 (Molecular Devices). One recording channel for the field electrophysiology was recorded with 1–2 MΩ borosilicate pipettes pulled the day of recordings and filled with the same aCSF in the recording chamber. The electrode was placed in the stratum pyramidale of the CA1 region at a depth of 20–25 um from the top of the slice as visualized under high power microscopy. This depth enabled optimal SWR detection without slice damage, and a standardized depth reduced variability across slices. Slices were binarily classified as having SWR or not if the experimenter could visually detect them at least 10 minutes after electrode placement in gap-free recordings. A final 2-minute-long recording from each slice in current clamp was performed at least 30 minutes after electrode placement to allow the slice to acclimate to the recording conditions.

## Data analysis

Sharp-wave and ripple events were automatically detected from 2-minute-long recordings using open-source MATLAB software which enables quantification of metrics including, but not limited to, rates, amplitudes, and durations of events in raw or filtered field potentials [22]. As previously described, this software detects SWR events by flagging all instances of lasting at least 10 milliseconds that are concurrently elevated greater than four standard deviations above baseline in both the sharp wave (1–30 Hz) and ripple (120–220 Hz) frequencies [22]. If SWR events are detected, those events are analyzed for metrics including duration (start and end times), power (area under the curve at the analyzed frequency), amplitude (baseline subtracted from sharp wave peak), and number of cycles (filtered only, e.g., ripple cycles) from start to end of SWR event [22]. When SWR were detected in multiple slices from the same animal, we averaged the data to generate a single datapoint per animal. The raw local field potential (LFP) files were batch analyzed using the swrAnalysis software with the settings outlined in Table 1.

## Statistics

All statistics were performed in the Python distribution of DABEST (https://acclab.github.io/DABEST-python/). Prior to statistical analysis, all data was checked for normality using the Shapiro-Wilk test. No data violations were found. A combination of estimation statistics and hypothesis-based significance were used with all data displayed as Cummings estimation plots [28]. We report all data with a categorical effect size and significance based on estimation statistics and student's t-test respectively. Categorical effect size was obtained from the calculated Cohen's d as either no effect (Cohen's $d < 0.5$), moderate effect (Cohen's $d > 0.5$), large effect (Cohen's $d > 0.8$), and very large effect (Cohen's $d > 1.2$) [28]. Significance-based testing was performed using unpaired t-tests. All datapoints and sample sizes represent number of animals. If one animal had multiple slices recorded from, the results from each slice were averaged into a single value displayed in the results. A table with all statistical values is provided to summarize the different tests used and their results in addition to the figures and text (Table 2). Outlier testing was performed with the ROUT test prior to analysis with a threshold of $Q > 1\%$. One animal from the HFHI group was statistically identified as an outlier in over half of the metrics reported and thus was removed from the entire study.

## Results

### SWR are grossly intact after HFHI

To assess SWR after HFHI, we prepared acute slices 24 hours after the last head impact in the HFHI protocol using male C57Bl/6 mice (Fig 1A). LFPs were measured in stratum pyramidale of the CA1 region in 500 um thick transverse

**Table 1. SWR analysis parameters.**

| | |
|---|---|
| **LFP filter** | 1-1000 Hz |
| **SW filter** | 1- 30 Hz |
| **Ripple filter** | 120-220 Hz |
| **RMS period** | 10 ms |
| **Baseline detection** | Gaussian fit |
| **Peak similarity** | 2 |
| **SWR window** | 100 ms |
| **Peak detection** | Concurrent SW and ripple |
| **Min. IEI** | 25 ms |
| **Min. Dur** | 10 ms |

LFP – local field potential. SW – Sharp wave. RMS – root mean squared. SWR – Sharp wave ripple. IEI – inter-event interval.

**Table 2. Summary of statistical results.**

| Metric | n Sham | n HFHI | Mean difference | 95% CI | Cohen's d | Effect size | 95% CI | p-value | t-statistic |
|---|---|---|---|---|---|---|---|---|---|
| SWR rate (Hz) | 8 | 7 | 0.05 | (−0.18, 0.20) | 0.26 | Small | (−0.97, 1.69) | 0.63 | −0.50 |
| SWR duration (ms) | 8 | 7 | −22.13 | (−45.56, −7.11) | −1.08 | Large | (−1.80, 0.05) | 0.06 | 2.09 |
| SWR amplitude (uV) | 8 | 7 | −4.42 | (−7.33, −1.54) | −1.39 | Very large | (−2.38, −0.29) | 0.02 | 2.69 |
| SWR power (uV2) | 8 | 7 | −96.54 | (−180.09, − 42.23) | −1.36 | Very large | (−2.16, −0.39) | 0.02 | 2.62 |
| Low gamma power (uV2) | 8 | 7 | −4.93 | (−9.10, 0.09) | −1.02 | Large | (−2.43, 0.25) | 0.07 | 1.97 |
| Low gamma cycles | 8 | 7 | −0.56 | (−1.23, −0.07) | −0.89 | Large | (−1.63, 0.24) | 0.11 | 1.73 |
| Low gamma phFreq (Hz) | 8 | 7 | 1.90 | (0.35, 3.76) | 1.05 | Large | (−0.09, 2.01) | 0.06 | −2.02 |
| Ripple power (uV2) | 8 | 7 | −13.98 | (−25.71, −2.33) | −1.12 | Large | (−2.15, 0.17) | 0.05 | 2.16 |
| Ripple cycles | 8 | 7 | −3.26 | (−5.94, −1.06) | −1.22 | Very large | (−2.07, −0.10) | 0.03 | 2.36 |
| Ripple phFreq (Hz) | 8 | 7 | 2.59 | (−3.35, 9.48) | 0.38 | Small | (−0.78, 1.49) | 0.48 | −0.73 |
| Fast ripple power (uV2) | 8 | 7 | −12.05 | (−23.37, −2.35) | −1.08 | Large | (−2.05, 0.07) | 0.06 | 2.09 |
| Fast ripple cycles | 8 | 7 | −6.95 | (−12.96, −2.00) | −1.17 | Large | (−2.03, 0.00) | 0.04 | 2.25 |
| Fast ripple phFreq (Hz) | 8 | 7 | 7.56 | (−1.5, 17.23) | 0.74 | Medium | (−0.37, 1.82) | 0.18 | −1.43 |

Metrics for effect size, mean difference and Cohen's d, along with their respective confidence intervals are shown. Categorical effect size was determined using thresholds for Cohen's d. Note, 8 animals were recorded from but one HFHI was identified as an outlier and removed from the statistical analysis (see methods).

slices. Prior to recording, a blinded experimenter classified slices as having the presence or absence of SWR through visual inspection of gap-free recordings. From an initial n = 11, the blinded reviewer did not detect SWR in recordings from 3 sham brains and from 3 HFHI brains, and detected SWR in 8 brains from each group (Fig 1B). In later statistical testing of the data, one HFHI brain failed outlier testing, leaving a final n = 8 Sham and n = 7 HFHI. The absence of SWR in some animals may be caused by variations in slice preparation and maintaining viable tissue. In the remaining brains, we used swrAnalysis software to identify, analyze average amplitude, rate, duration, and power of SWR events from gap-free recordings. Raw LFP, sharp waves, ripple events, and spectrograms are shown in Fig 1C, and 100 ms traces windowed around detected events are shown in Fig 1D. Events from HFHI animals show grossly intact SWR events with an expected waveform and frequency from stratum pyramidale [20]. These data show that SWR are present in both sham and HFHI brains.

## HFHI alters hippocampal SWR

We quantified features of SWR from automatically detected events in gap-free recordings and compared between experimental groups. SWR recorded from HFHI animals showed a significant decrease in amplitude with a very large effect size (Fig 2A; mean difference, −4.42; 95% CI, −7.33, −1.54; Cohen's d, −1.39; p-value, 0.02; t-statistic, 2.69) but no change in the rate of events (Fig 2B). There was no effect on the SWR event duration in HFHI animals (Fig 2C), however there was a significant reduction in total SWR power (area under the SWR curve) in HFHI animals with a very large effect size (Fig 2D; mean difference, −96.54; 95% CI, −180.09, −42.23; Cohen's d, −1.36; p-value, 0.02; t-statistic, 2.62). These data show that SWR events are generated in the HFHI brain, but events are altered with respect to amplitude, duration, and overall power.

## HFHI reduces ripple power

Ripples are fast oscillations between 120−220 Hz that occur in the CA1 region. They may occur with or without coinciding sharp wave events and it is thought that the depolarization that occurs during sharp waves opens a window for ripple events to occur [20,26,27]. Phase-locked firing during ripple events carries information for memory consolidation

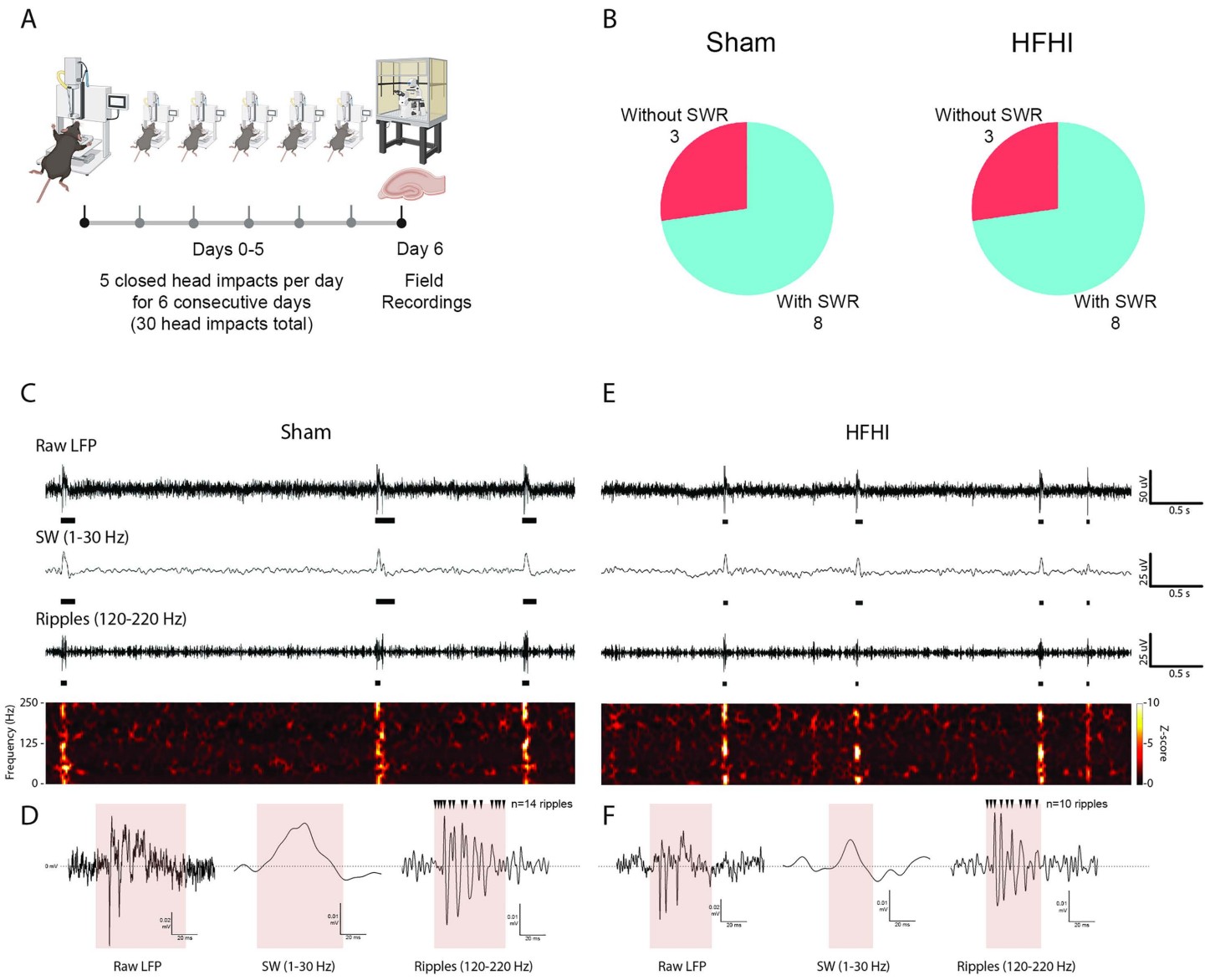

**Fig 1. Experimental design. A)** Experimental timeline. Male WT mice were given 30 head impacts or equivalent isoflurane and handling over the course of 6 days. 24 hours after the last day of impacts acute slices were prepared for field electrophysiology. **B)** Pie charts showing number of animals from sham (left) and HFHI (right) groups with visually detected SWR. **C)** Example 5 second trace from sham slice of LFP, SW (1-30 Hz), ripples (120-220 Hz), and a z-scored spectrogram (1-250 Hz). **D)** Corresponding plots within a 100 ms windows around a SWR event for that slice. **E)** Example 5 second trace from HFHI slice of LFP, SW (1-30 Hz), ripples (120-220 Hz), and a z-scored spectrogram (1-250 Hz). **F)** Corresponding plots within a 100 ms windows around a SWR event for that slice.

and planning and are generated by fast-spiking interneurons in the CA1 region [26,29]. We quantified features of ripples (120−220 Hz) during detected SWR events. There was a statistically significant decrease in ripple cycles with a very large effect size (Fig 3A; mean difference, −3.26; 95% CI, −5.94, −1.06; Cohen's d, −1.22; p-value, 0.03; t-statistic, 2.36) in HFHI mice compared to shams, however, there was no change in peak ripple frequency (Fig 3B, mean difference, 2.59; 95% CI, −3.35, 9.48; Cohen's d, 0.38; p-value, 0.48; t-statistic, −0.73). Ripple power (area under the curve) measured during SWR-locked events showed a statistically significant decrease with a large effect size (Fig 3C, mean difference,

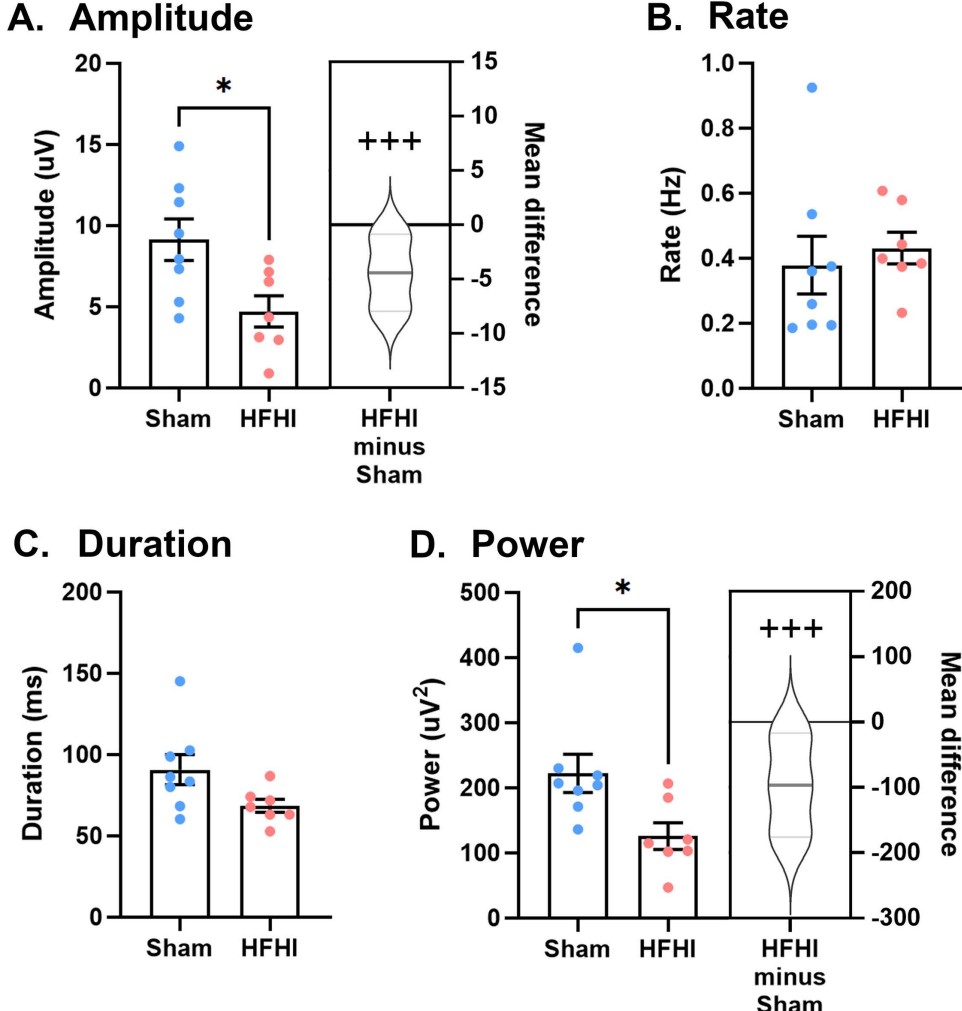

**Fig 2. Sharp wave architecture in sham and HFHI mouse brain A) SWR amplitude, B) rate of events, C) duration, and D) power.** N = 8 for sham and n = 7 for HFHI. Bar graphs show mean +/- SEM, with each datapoint representing an individual mouse. * p < 0.05 students t-test. Cummings estimation plots show mean difference as HFHI – Sham and half violin of Monte-Carlo bootstrapped data with 5000 permutations. +++ = very large effect size (Cohen's d > 1.2).

--13.98; 95% CI, −25.71, −2.33; Cohen's d, −1.12; p-value, 0.05; t-statistic, 2.16). These data show that ripple features in the HFHI SWR events show decreased number of cycles and power corresponding to both the duration and amplitude.

### Minimal detectable changes in pathologic or gamma rhythms

While the predominant phenotype of SWR events occur in the sharp wave (1−30 Hz) and ripple (120−220 Hz) frequency bands, other known phenotypes like fast ripples (250−500 Hz) and low-gamma (20−50 Hz) co-occur with these events and may indicate pathologic rhythmogenesis and increased memory performance respectively [20,22,29]. We quantified the cycles, peak frequency, and power for both fast ripples and low-gamma bandwidths [22,29]. Recordings from HFHI showed a significant decrease in number of fast ripple cycles with a large effect size (Fig 4A; mean difference, −6.96; 95% CI, −12.96, −2.00; Cohen's d, −1.17; p-value, 0.04; t-statistic, 2.25), but no change in fast ripple frequency (Fig 4B) or power (Fig 4C). No changes to low-gamma cycles, frequency or power were observed (Fig 4D-4F). Taken together,

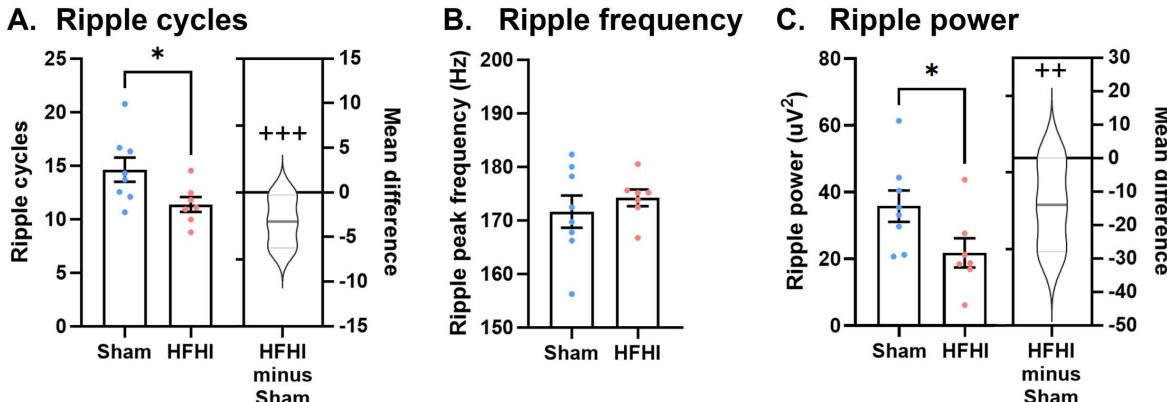

**Fig 3. Ripple features during sharp waves in sham and HFHI brain.** Quantification of ripple architecture in sham and HFHI mouse brain: **A)** ripple cycles, **B)** ripple frequency, and **C)** ripple power. N = 8 for sham and n = 7 for HFHI. Bar graphs show mean +/- SEM, with each datapoint representing an individual mouse. * p < 0.05 students t-test. Cummings estimation plots show mean difference as HFHI – Sham and half violin of Monte-Carlo bootstrapped data with 5000 permutations. ++ = large effect size (Cohen's d > 0.8), +++ = very large effect size (Cohen's d > 1.2).

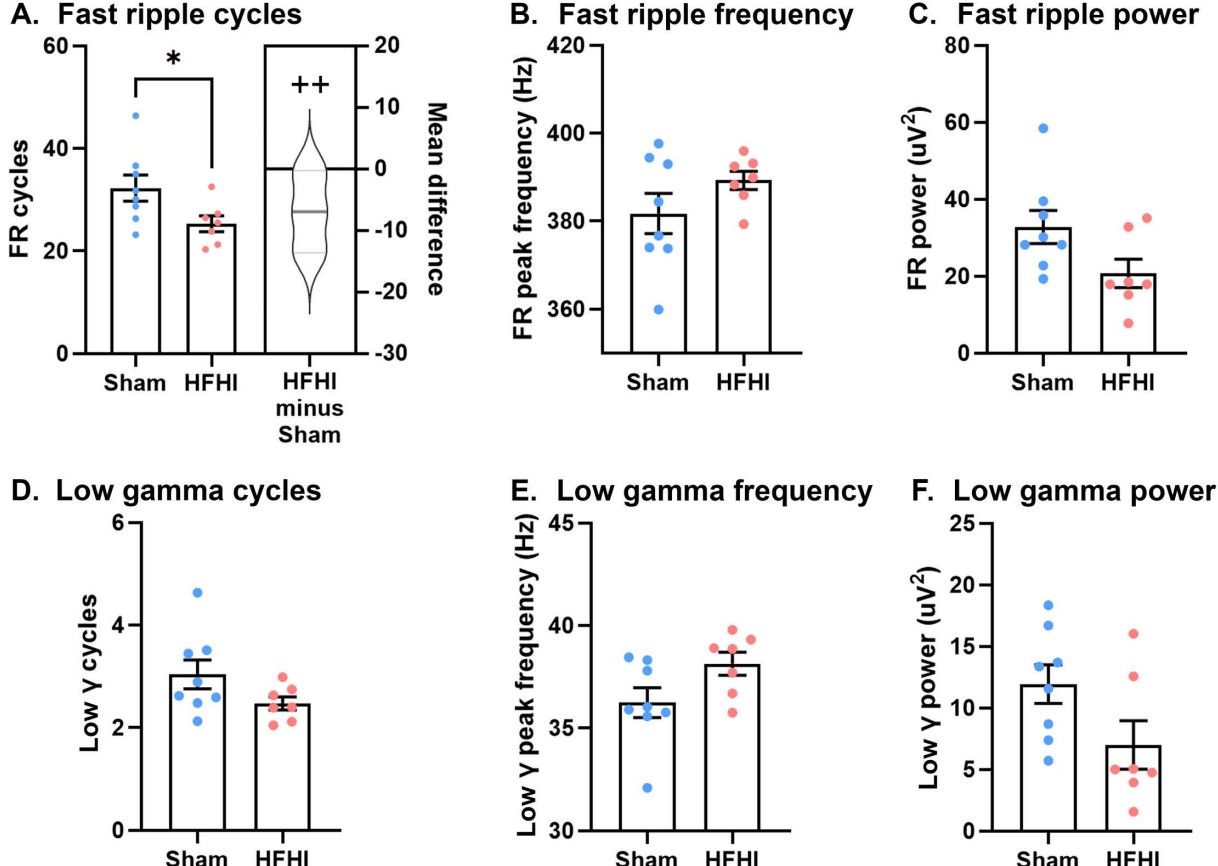

**Fig 4. Characterization of fast ripples and gamma during SWR.** **A)** fast ripple cycles, **B)** fast ripple frequency, and **C)** fast ripple power. **D)** Low-gamma cycles, **E)** low-gamma frequency, and **F)** low-gamma power in SWR locked events. N = 8 for sham and n = 7 for HFHI. Bar graphs show mean +/- SEM, with each datapoint representing an individual mouse. * p < 0.05 students t-test. Cummings estimation plots show mean difference as HFHI – Sham and half violin of Monte-Carlo bootstrapped data with 5000 permutations. ++ = large effect size (Cohen's d > 0.8).

our data shows no significant changes to low-gamma rhythms and a reduction in pathological ripples cycles during SWR events.

## Discussion

The present study measures and characterizes the spontaneous occurrence of hippocampal SWR in a mouse model of the repeat head impact brain. By measuring SWR in acute slices 24 hours after a week-long head impact protocol, we show that spontaneous SWR are present and grossly intact after repeated head impacts but show decreased amplitude and power with blunted rhythms in canonical ripples (120–220 Hz).

SWR have been extensively characterized in murine models and more recently recorded in human hippocampus [20,30]. Sharp waves are generated by building excitation in the recurrent CA3 network and result in depolarization of the CA1 apical dendrites [20,26]. The depolarization creates a window for co-occurring ripple oscillations which are generated through fast spiking by local parvalbumin positive basket cells [20,26]. These events occur during consummatory behavior such as planning and sleep *in vivo* and occur spontaneously in the acute slice preparation when thicker (generally >450 um) transverse slices are used [20,30]. The presence of SWR during slow-wave sleep is strongly associated with memory performance and their disruption dramatically reduces cognitive function while electrographically, hippocampal SWR are associated with cortical spindles and slow wave activity [24,31,32]. Deficits to synaptic communication, neuronal excitability, and circuit architecture are known to disrupt normal features of SWR *in vivo* and in slice and are often accompanied by severe cognitive deficits [22,33,34]. Discernable features of SWR (i.e., amplitude, duration, rate, etc.) in either the sharp wave and/or ripple component have been linked to unique neuronal mechanisms and functional outcomes [23,35,36]. In the present study, we found sharp wave and co-occurring ripples 24 hours post-injury in HFHI mice and sham controls in acute slices. Previous characterization of the HFHI model shows that this head trauma model does not cause hippocampal pathology, inflammation, or neuronal cell death [15,16], however HFHI mice do have hippocampal synaptic dysfunction and cognitive deficits. The addition of the current data showing that spontaneous SWR exist in the HFHI brain, but have altered architecture supports the concept that the cognitive deficits in HFHI mice are caused by altered synaptic and circuit physiology rather than TBI pathology.

Previous studies of SWR in TBI are sparse. Two studies in more severe models of TBI have also reported the presence of SWR. One study recorded SWR in rats 6–11 days after lateral fluid percussion injury and found that rate of events was unchanged compared to control animals [37]. Another study recorded SWR 28–35 days after an air blast injury and found decreased SWR rate [38]. Both models are single injury models and are known to cause hippocampal neuron cell death and pathology [39,40], which is not seen in the HFHI model. Taken together, this suggests that gross SWR generation is not broadly impaired after either mild/moderate TBI or high-frequency head impacts although their presence/absence in more severe injury models like controlled cortical impact is currently unknown. The variability in rate across the studies is likely specific to injury mechanism and may represent the functionality of the recurrent CA3 and CA3-CA1 networks, or inhibitory control, in different TBI models. The presence/absence and rate of SWR events does not alone predict memory impairment.

Despite grossly normal events, HFHI mice showed a decreased SWR power affected by both amplitude and duration of events. Similarly, in a 5xFAD Alzheimer's model, decreased duration was accompanied by decreased ripple cycles which likely contributed to the shorter duration and decreased ripple power [22]. There is causal evidence that longer duration SWR events correlate with improved memory [23], suggesting that the decreased SWR duration could be contributing to impaired memory performance in the HFHI model [16,18]. In the lateral fluid percussion model, ripple duration was observed to be increased compared to control animals [37] while the air blast model showed decreased peak ripple frequency [38]. This shows a divergence in phenotype between more severe models of TBI and our HFHI model. This may give us more insight into circuit mechanisms as ripples are generated by inhibition and more severe models of TBI show increased excitation/inhibition (E/I) ratio likely due to decreased inhibition and not increased excitation [27,41–43].

Contrarily, this could be explained by hyperexcitability in the principal cells of the CA3 and CA1 regions that produce a longer depolarization in the low frequency sharp wave bandwidth, keeping the window for ripple events to occur open longer.

In both Alzheimer's and the more severe models of TBI it is postulated that selective disruption to inhibitory neurons through either cell death or pathology-dependent pathways underlies the observed effects on SWR [22,43]. PV interneurons are substantially decreased after severe experimental brain injury in young mice [44] and transplanting GABAergic precursors to the lesion following severe experimental brain injury improves memory performance [45]. Lateral fluid percussion, but not air blast, reduced the amplitude seen in SWR [37,38]. The decreased power of both SWR and isolated ripples seen in HFHI mice is also reflected in the lower amplitude of SWR in the same mice. The mechanism by which this happens in HFHI is not known although we hypothesize it is secondary to the diffuse glutamatergic synaptic dysfunction previously recorded as this model does not lead to neuron cell death like the PV-interneuron loss seen after severe experimental brain injury.

Epileptiform activity and seizures are brain states that represent excessive synchronization of neural activity. Severe models of TBI like the controlled cortical impact and lateral fluid percussion models, in addition to evidence from humans, have shown increased propensity for epileptic seizures after brain injury [46,47]. We have not observed seizures in the HFHI model to date and mild TBI in humans is not strongly associated with epileptic activity even in repeat incidences [48]. No significant change was seen in fast ripple cycles, amplitude, or power in HFHI mice. Previous studies measuring fast ripples in TBI are sparse. One study in an *ex-vivo* model of TBI in which the entorhinal cortex is cut prior to excision from the skull showed near-immediate emergence of fast ripple events [49].

The low gamma bandwidth (20–50 Hz), which is thought to correspond to memory replay in the CA3 and CA1 regions during SWR, also showed no significant change in HFHI mice [29]. We did not measure spike content or correlating theta rhythms in the present study, but studies in the lateral fluid percussion model have shown disruption to gamma coupling and synchronization to oscillations in the CA1 [37,50]. Similar studies done in pig slices after mild TBI (mTBI) also reveal decreased synchrony of gamma oscillations and hyperexcitability in the CA1 region [51,52]. We have previously shown decreased hippocampal ensemble coordination after HFHI [17]. Thus, we hypothesize that HFHI models, like more severe models of mTBI, has decreased synchrony in core memory circuits despite intact initial generation of normal brain rhythms.

Previous work in the lateral fluid percussion model found a broadband decrease in hippocampal power [53]. These results were obtained through *in vivo* experimentation and analyzed continuous data during theta-dominant exploration while the current study analyzed data *ex vivo* and only during detected SWR events. While we saw trends that suggest a broadband decrease in hippocampal power, this did not reach statistical significance so this remains a hypothesis with the current sample size. We did however, observe a decrease in fast ripple cycles in HFHI mice. Despite the differences in methodology, these results show a convergence of physiologic phenotype between the HFHI model during SWR events and theta-dominated exploration in the lateral fluid percussion model. Such results could serve as a biomarker for cognitive dysfunction post-TBI or repeat head impacts and help stratify injury across models based on the occurrence of power disruption during various brain states.

Beyond preclinical work, TBI is stratified into three categories based on purely clinical findings [2]. Recent advancements in blood-based biomarkers have aided in detection of injury in isolated events of TBI beyond clinical criteria, but cognitive dysfunction after repeat head impacts still lacks a formal clinical definition or criteria for diagnosis. Due to the extremely mild nature of some sports related head impacts, they are often missed, and cognitive decline may occur over the course of years rather than after any one event [2]. These features of sports-related head trauma make it difficult to detect early. Establishing electrographic biomarkers such as broadband power depression would be beneficial for finding such chronic injury to inform return-to-play decisions as well as treatment efficacy. The limitations of SWR-based approaches are lack of access to high fidelity recordings in the human hippocampus. Contrarily, SWR-like events have been detected in cortical areas of humans and may be detectable through extracranial recordings [54]. *In vivo* sleep

studies in the HFHI show no differences in broadband power during full sleep/wake cycles [55] but full analysis of cortical rhythms during cognitive behaviors has yet to be performed.

The current study was done in acute slices 24 hours after the last injury, and we acknowledge limitations to this study. First, the recordings were done *in vitro*. Despite using thick slices to preserve hippocampal network architecture, there are differences between *in vitro* and *in vivo* SWR [20]. Second, further studies are needed at multiple timepoints to assess whether the changes to spontaneous SWR observed here are chronic or if they spontaneously recover over weeks to months. The retrograde and anterograde memory impairments are seen chronically but the intrinsic hypoexcitability of CA1 neurons is short lived and gene expression is dynamic [15,16,18]. The current study does not illuminate the time course of SWR changes after HFHI. Third, only field recordings were used and therefore no mechanistic insights can be gained.

To date, no FDA approved treatments for TBI or repeat head impacts exist. In addition to serving as a cognitive biomarker, modulation of SWR has been proposed as a possible treatment option for memory disorders. Evidence for other targets of deep brain stimulation such as the thalamus has shown promise in treating moderate to severe TBI [56]. Given that SWR genesis is not impaired in HFHI, but rather there is a broadband power decrease in SWR locked events in the setting of decreased duration and amplitude, a closed loop recording-stimulation approach that extends and amplifies naturally generated SWR would be preferred. Artificial generation and pacing of SWR has sh own little success in other domains of memory improvement compared to online stimulation and would not address the altered SWR features after injury. The major limitation to this approach is the accessibility and invasiveness of depth electrodes in the hippocampus. A separate approach would be to address the broadband power decrease in the hippocampus. As mentioned above, it remains to be seen whether this power decrease is limited to SWR events in the hippocampus, but previous EEG recordings after HFHI would suggest cortical power remains unchanged [55]. As non-invasive neuromodulation technologies achieve greater spatial resolution, it may become possible to selectively excite the hippocampus which is otherwise depressed in the HFHI model.

## Author contributions

**Conceptualization:** Daniel P. Chapman, Stefano Vicini, Mark P. Burns.

**Data curation:** Daniel P. Chapman.

**Formal analysis:** Daniel P. Chapman, Margaret S. Sten.

**Funding acquisition:** Daniel P. Chapman, Mark P. Burns.

**Investigation:** Daniel P. Chapman, Margaret S. Sten.

**Methodology:** Daniel P. Chapman.

**Project administration:** Mark P. Burns.

**Resources:** Stefano Vicini, Mark P. Burns.

**Supervision:** Stefano Vicini, Mark P. Burns.

**Writing – original draft:** Daniel P. Chapman, Mark P. Burns.

**Writing – review & editing:** Daniel P. Chapman, Margaret S. Sten, Stefano Vicini, Mark P. Burns.

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
