## [Decision Letter · Decision Letter 0]

14 May 2025

Dear Dr. Burns,

Thank you for submitting your manuscript to PLOS ONE. After careful consideration, we feel that it has merit but does not fully meet PLOS ONE’s publication criteria as it currently stands. Therefore, we invite you to submit a revised version of the manuscript that addresses the points raised during the review process.

The reviewers are generally favorable, with some suggested edits to the manuscript and requests for clarification. Please respond carefully to the reviewers, addressing each point.

We look forward to receiving your revised manuscript.

Kind regards,

Julian Cheron

Academic Editor

PLOS ONE

Journal Requirements:

“Research reported in this publication was supported by National Institute of Neurological Disorders and Stroke of the National Institutes of Health under award number R01NS107370, R01NS121316 & F30NS122281.  The content is solely the responsibility of the authors and does not necessarily represent the official views of the National Institutes of Health.”

Reviewers' comments:

Reviewer's Responses to Questions

**Comments to the Author**

1. Is the manuscript technically sound, and do the data support the conclusions?

Reviewer #1: Yes

Reviewer #2: Partly

2. Has the statistical analysis been performed appropriately and rigorously?

Reviewer #1: Yes

Reviewer #2: Yes

3. Have the authors made all data underlying the findings in their manuscript fully available?

Reviewer #1: Yes

Reviewer #2: Yes

4. Is the manuscript presented in an intelligible fashion and written in standard English?

Reviewer #1: Yes

Reviewer #2: Yes

Reviewer #1: Paper by Chapman et al. (submission to PLoS One)

High frequency head impact exposure changes hippocampal sharp-wave ripple architecture

Summary

This manuscript explores the relationship between head injury and hippocampal dynamics, with in vitro measurements in mice. This manuscript’s main question: how do repeated head impact change oscillatory patterns in the hippocampus CA1 region, and more precisely how repeated impacts affect sharp waves and ripples? The mode of injury is described as high-frequency head impact (HFHI), which has low impact force, but presents the effect of multiple cumulative small impacts, such as in football/soccer, or other action sports. The authors have published using this HFHI method previously. After the experimental manipulation, the authors record field potential activity in the hippocampus CA1 and compare the changes in the spike-wave and ripple combo activity, along with the activity of the fast ripples, and gamma periods. Across these measurements, it appears like HFHI mice show lower power of activity across these frequency boundaries. Thus, overall, the authors propose an interesting study, with results that help describe fast oscillatory components that change with injury. While speaking more as a generalist of field potentials, the methods appear well executed, and in general the data acquisition and analysis also seem to be well done, though I have a few questions. Overall, the discussion and conclusions appear sound. This paper has the potential to solidly advance our knowledge in the neurophysiological changes emanating from HFHI.

Major points:

Results

1. Line 184: it would help to see certain examples of data reduction - so how the data gets to be reduced to one data point per animal. A figure (a histogram, for example) showing how the data from multiple recording periods gets to be reduced to one value. This would also provide context as to the variability of measurements per animal.

2. Between lines 197 and 202: what was it, visual inspection or automated detection? While I’m confident the authors used a reliable algorithmic approach to the detections, the explanations “flip” a few times between the “visual inspection” and “automatic detection”. This deters from the comprehension.

3. Figure 1D: what is the meaning of the zoomed-in spectrogram? Unless this carries specific information – this can be removed.

4. Line 220: the SWR power should be defined more clearly - is it the spike wave amplitude, or is it the area under the “SWR curve”? This explanation would provide more clarity as to the power being measured.

5. Line 232 and figure 3: it appears that the ripples are 10-12 cycles long… this seems very long compared with the examples in Figure 1 which are 5-6 cycles. This should be established – maybe with a sample trace next to the figure?

6. Similar to #4, but here likely more straightforward… Figure 3C presents ripple power is it the area under the spectrogram curve between 120 Hz and 220 Hz?

7. Line 245: the authors have explained the selection of their gamma frequency band between 20 Hz and 50 Hz: while literature is not always fixed on the frequency, some examples give low gamma between 30-70 Hz and high gamma between 70-150 Hz (e.g., Catanese et al., 2016, JNeurophysiol). Without advocating for change, at least the authors should explain their choice.

Minor points:

Abstract

• Line 51: what does it mean in the absence of pathology? Anatomical disturbance?

Introduction

• line 68: it would help if examples of observable pathology were provided (lesions, cortical thinning?)

Methods

• line 160: The authors should provide more information concerning the selection of the depth - was this to capture the best LFPs?

• line 166: please clarify the order of the first two sentences in “Data analysis” - it might be better to flip them?

Results

• Line 200: is it really an artifact? More simple word – a cause?

• Line 215 and figure 2B: the terminology is equivocal - is it frequency or rate of SWR events? Rate seems better – it feels like frequency could be confusing as it is used to describe the oscillatory phenomena.

• Lines 220 to 228: While this might be a specific personal style, there is usually no need to provide description and citations here… this should be in the introduction or discussion; same goes for lines 243-246… this should be checked re: the journal’s style.

• Figure 4D: The Y axis should not have “gamma ripple” - this seems like a mistake.

• Line 336: CCI should be defined.

Reviewer #2: In their manuscript, Chapman et al. investigate the impact of high-frequency head impacts (HFHI) on hippocampal function in mice, focusing on sharp-wave ripples (SWR), a known biomarker of memory and cognition. The authors demonstrate that although SWRs remain structurally intact post-HFHI, their amplitude, power, and ripple cycle count are significantly reduced compared to controls. These findings suggest that HFHI impairs intrinsic hippocampal network activity, potentially contributing to cognitive deficits observed in the absence of overt brain pathology. The work provides interesting insight into subtle neurophysiological changes following repetitive head trauma

I have a few comments and suggestions regarding the work listed as they appear throughout the manuscript.

- Abstract:

The notion of ‘reduced architecture’ in the last sentence of the abstract feels imprecise. I suggest the authors use a different terminology

- Introduction:

The last sentence of the introduction (lines 105-106) should include that the work was performed on acute braine slices

- Methods:

Electrophysiology

Lines 161-163, It’s not totally clear how many 2-min-long recordings were performed on each slice.

It could be useful to also provide the average number of slices recorded per animal and the average number of recordings performed on each slice.

Data Analysis

The authors should define in a few words the outputs from the Matlab software they quantified

In line 171, the authors mention the ‘best 2-minute recording’. What are the actual criteria used to sort or classify recording quality?

Statistics

Prior to the use of parametric tests, did the authors check for normality and homoscedasticity? In case if violation, did the authors perform corrections?

- Results:

In line 198, because the authors removed one outlier animal in the HFHI group, shouldn’t it be 7 out of 10 for the HFHI group?

The panel A in Figure 1 may be slightly revised to make it more obvious that the head impacts protocol occurred during 6 consecutive days, for example by showing the 6 different points corresponding to each of the 6 days.

In Figure 1, panels D and F, the Y axis for the spectrograms are missing.

As mentioned above regarding the methods, the features extracted from SWR events should be more clearly explained. For example, in lines 216-217, about the quantification of SWR duration, the authors refer to the text in the methods section, but there is no detailed information in this section. This additional information should be provided directly in the manuscript. Some schematic panels illustrating the extracted features may be also added in Figures 2-4

Some of the results reported by the authors fall very short on reaching the threshold for significance while displaying large Cohen effect sizes (eg SWR duration, gamma power, fast ripple power). Thus, the authors should consider increasing the sampling size to strengthen their results.

- Discussion:

Line 326, I may be mistaken, but it looks like ‘CCI’ was not defined before.

Lines 351-353, The authors should more explicitly state that this statement is a hypothesis.

Line 354, As mentioned above, this lack of statistical significance may result from a low sampling.

**Do you want your identity to be public for this peer review?** For information about this choice, including consent withdrawal, please see our Privacy Policy

Reviewer #1: **Yes: ** Richard Courtemanche

Reviewer #2: No

---

## [Author Response · Author response to Decision Letter 1]

17 Oct 2025

Reviewer Response is attached as a separate file

---

## [Editor Report · Decision Letter 1]

4 Nov 2025

High frequency head impact exposure changes hippocampal sharp-wave ripple architecture

PONE-D-25-06006R1

Dear Dr. Burns,

We’re pleased to inform you that your manuscript has been judged scientifically suitable for publication and will be formally accepted for publication once it meets all outstanding technical requirements.

Kind regards,

Julian Cheron

Academic Editor

PLOS ONE
---

## [Editor Report · Acceptance letter]

PONE-D-25-06006R1

PLOS One

Dear Dr. Burns,

I'm pleased to inform you that your manuscript has been deemed suitable for publication in PLOS One. Congratulations! Your manuscript is now being handed over to our production team.

Kind regards,

on behalf of

Dr. Julian Cheron

Academic Editor

PLOS One